# The Impact of Hypertension, Diabetes, Lipid Disorders, Overweight/Obesity and Nicotine Dependence on Health-Related Quality of Life and Psoriasis Severity in Psoriatic Patients Receiving Systemic Conventional and Biological Treatment

**DOI:** 10.3390/ijerph182413167

**Published:** 2021-12-14

**Authors:** Anna Karpińska-Mirecka, Joanna Bartosińska, Dorota Krasowska

**Affiliations:** 1Department of Dermatology, Venereology and Pediatric Dermatology, Medical University of Lublin, 20-081 Lublin, Poland; dor.krasowska@gmail.com; 2Department of Cosmetology and Aesthetic Medicine, Medical University of Lublin, 20-093 Lublin, Poland; jbartosinski@gmail.com

**Keywords:** plaque psoriasis, HRQOL, DLQI, PASI, comorbidities, nicotine dependence, overweight/obesity, conventional and biologic treatment

## Abstract

Psoriasis, a chronic disease, is associated with a higher prevalence of comorbidities and has negative impact on health-related quality of life (HRQOL). The objective was to investigate the effect of comorbidities on HRQOL, and psoriasis severity measured appropriately by the dermatology life quality index (DLQI) and the psoriasis area severity index (PASI) before, and after a 3-month treatment and the median DLQI or PASI reduction from baseline in the adult psoriatic patients receiving various types of treatment. The study included 184 adult plaque psoriatic patients. DLQI and PASI scores were assessed in the studied patients before the therapy (a baseline visit) and after a 3-month treatment (a control visit) depending on the presence of comorbidities. Psoriatic patients with comorbidities had worse HRQOL and more severe skin lesions. The presence of comorbidities had a negative effect on the outcome of treatment with the use of conventional therapy. The outcome of therapy with biological agents was independent of each of the analyzed factors. Biological treatment had a high effectiveness on the psoriatic skin lesions improvement despite the presence of comorbidities, whereas methotrexate was effective even if the patients had co-existing hypertension. In psoriatic patients receiving systemic conventional treatment but not biological treatment, comorbidities had a negative impact on HRQOL and psoriasis severity.

## 1. Introduction

Psoriasis is a chronic autoinflammatory skin disease affecting about 1–4% of the world population [1]. Apart from its debilitating effect on the patient’s physical health, psoriasis disturbs the patient’s mental well-being, interferes with their daily routines and work activities as well as social and family relationships [2,3,4]. The health-related quality of life (HRQOL) in psoriatic patients is most often measured and assessed with the use of the dermatology life quality index (DLQI) which consists of 10 questions concerning the patient’s daily routines, social and professional activities as well as their feelings and experiences [5,6]. Each question is given from 0 to 3 points, and the maximum number of points to score in the DLQI questionnaires 30 [7]. The higher the score, the worse the patient’s quality of life [8].

In order to measure the severity of psoriasis, psoriasis area and severity index (PASI), body surface area (BSA) and dermatology life quality of index (DLQI) are used. The psoriasis area and severity index score is a measurement of the erythema, thickness, scaling and coverage of the plaques. The body surface area is the factor which measures the area of the skin is covered by psoriasis. The result is given as a percentage [9]. On the basis of the PASI score, BSA and DLQI patients have been stratified into three groups: mild disease, moderate and severe psoriasis. Moderate to severe psoriasis was diagnosed when BSA ≥ 10%, and/or PASI ≥ 10, and/or DLQI ≥ 10 [10].

More often than not, psoriatic patients suffer from cardiovascular diseases, metabolic disorders, i.e., diabetes, overweight/obesity and lipid disturbances [11,12]. Undoubtedly, smoking exacerbates the course of psoriasis, has a negative effect on the outcome of treatment, and it is conducive to serious complications [13].

The aim of the study was to investigate the impact of comorbidities, overweight/obesity and nicotine dependence on HRQOL measured by the DLQI prior to and after a 3-month treatment. The study also aimed at assessing the median DLQI reduction from baseline in the adult psoriatic patients receiving various types of systemic treatment, both conventional and biological, and investigating PASI prior to and after a 3-month treatment as well as the median PASI reduction from baseline and PASI score improvement in the psoriatic patients with comorbidities, overweight/obesity and nicotine dependence treated either conventionally or biologically [14,15].

## 2. Materials and Methods

### 2.1. Characteristics of the Studied Group

The study was conducted between May 2019 and October 2020, and it included 184 patients with plaque psoriasis (97 males and 87 females), mean age 46 ± 12 years (range 20–80 years). The duration of the disease was15 ± 10 years, on average.

The exclusion criterion was the age under 18 years. All the patients fulfilled the criteria of moderate to severe plaque psoriasis, i.e., in all of them, at least one of the indices (PASI, BSA or DLQI) was 10 or higher.

All the studied subjects were treated with either conventional systemic drugs, i.e., methotrexate (42.9% of the patients), cyclosporine (22.8% of the patients), retinoids (23.4% of the patients), or biological agents, i.e., adalimumab, ustekinumab and secukinumab (10.9% of the patients). Hypertension, diabetes and lipid disorders were diagnosed in 62, 40 and 60 of the studied patients, respectively. Out of the total number of the studied subjects, 65 of them were smokers. The detailed clinical and demographic characteristics of the studied patients are presented in Table 1. This study was approved by the ethic committee of the Medical University of Lublin (No KE-0254/154/2019).

All the studied psoriatic patients were interviewed, their medical history was taken, and they underwent physical examination. Severity of psoriasis was assessed with the use of the psoriasis area severity index (PASI), body surface area (BSA) and the dermatology life quality index (DLQI) [16,17,18,19]. Moderate to severe psoriasis was diagnosed when BSA ≥ 10%, and/or PASI ≥ 10, and/or DLQI ≥ 10 [10]. In each patient, BMI was calculated as the body mass in kilograms divided by the squared height in metres [20]. When BMI was ≥25, the patient was regarded as overweight/obese [21]. A total of 50% of the patients were overweight or obese. In 92 patients, the calculated body mass index (BMI) was 25 or higher.

The DLQI and PASI scores were assessed prior to launching therapy (a baseline visit) and after a 3-month treatment (a control visit) in all of the studied psoriatic patients, and in the patients treated with different therapy regimens depending on the presence or absence of comorbidities, BMI score and smoking status.

A 3-month treatment is a period long enough to evaluate the first effect of treatment and improvement in quality of life. Many conventional and biological drugs are evaluated after a 3-month period.

The median DLQI and PASI reductions from baseline were also calculated. The numbers and percentages of the patients achieving improvement in the PASI score, i.e., PASI50, PASI75, PASI90, after a 3-month treatment were analyzed with reference to the presence or absence of comorbidities, BMI score and smoking status.

### 2.2. Statistical Analysis

Statistical analysis was conducted with the use of the MedCalc computer software, version 18.2.1 (MedCalc, Ostend, Belgium). Data distribution in the studied population was examined by the Shapiro–Wilk test. All the studied parameters demonstrated abnormal distribution of the data (*p* < 0.05) therefore, non-parametric tests were applied in the further analysis. Multiple regression analysis for the clinical-demographic features of the studied individuals was applied to avoid the potential confounding effects on the results. None of the demographic factors affected either DLQI or PASI score, thus the whole studied group underwent non-parametric testing. Values of the studied factors in the subgroups of patients treated with different therapy regimens were expressed by the median (Me) and interquartile range (IQR). Differences in the values of the studied parameters among the patients in regard to the presence of comorbidities, BMI score and smoking status were tested by the Mann–Whitney U test. The results presenting *p* values below 0.05 were considered as statistically significant.

## 3. Results

### 3.1. DLQI Score in the Psoriatic Patients Prior to and after a 3-Month Treatment as well as the Median DLQI Reduction from Baseline Depending on the Presence or Absence of Comorbidities, BMI Score and Smoking Status

Comparison of the median DLQI scores in psoriatic patients treated with different therapy regimens depending on the presence of comorbidities (hypertension, diabetes, lipid disorders), BMI score and smoking status is presented in Table 2.

Prior to launching any of the applied therapies, the median DLQI scores at baseline were significantly different between the studied subjects with comorbidities, BMI ≥ 25 kg/m^2^ or smokers and the studied psoriatic patients presenting no aforementioned comorbidities, whose BMI was below 25 kg/m^2^, and also in non-smokers. The psoriatic patients with comorbidities (hypertension, diabetes, lipid disorders) or individuals with BMI ≥ 25 kg/m^2^ as well as smokers had significantly higher DLQI scores in comparison to the patients without comorbidities, whose BMI was below 25 kg/m^2^, and also in non-smokers (*p* = 0.004, *p* = 0.012, *p* = 0.014, *p* < 0.001, *p* = 0.006, respectively). Before the methotrexate therapy, the median DLQI scores were significantly higher in the patients with comorbidities, with BMI ≥ 25 kg/m^2^ and in smokers in comparison to the patients without comorbidities, with BMI < 25 kg/m^2^, and in non-smokers (*p* < 0.001, *p* < 0.001, *p* < 0.001, *p* < 0.008, respectively). In addition, before the treatment with cyclosporine or retinoids, the median DLQI scores were significantly higher in the patients suffering from diabetes (*p* = 0.014 and *p* = 0.027, respectively).

After a 3-month treatment, significant differences were still observed in the median DLQI scores between the studied subjects with comorbidities, BMI ≥ 25 kg/m^2^ or smokers and the studied psoriatic patients presenting no comorbidities, whose BMI was below 25 kg/m^2^ as well as in non-smokers. However, they were lower in comparison with the median DLQI scores from before the treatment. As for the methotrexate therapy, after a 3-month treatment the differences were still significant (*p* < 0.001). In the case of cyclosporine therapy, the presence of diabetes and BMI ≥ 25 kg/m^2^ negatively affected the median DLQI scores, while in the patients without diabetes and with BMI < 25 kg/m^2^ the median DLQI scores were significantly lower (median DLQI score 10.0 vs. 6.0, *p* = 0.003; 10.0 vs. 6.0, *p* = 0.021, respectively). In the patients treated with retinoids for 3 months, the median DLQI scores were significantly higher in the studied subjects with hypertension or diabetes as well as in smokers in comparison to the psoriatic patients without hypertension or diabetes as well as in non-smokers. None of the studied factors influenced the median DLQI scores in the patients receiving a 3-month treatment with biological agents.

The median DLQI reduction from baseline (i.e., from before launching any type of therapy to the control visit after 3 months) was analyzed in the studied psoriatic patients who received different treatment depending on the presence of comorbidities (hypertension, diabetes, lipid disorders), BMI score and their smoking status. The most significant differences were observed in the patients treated with biological agents. The median DLQI reduction was significantly higher in the non-smokers (14.0 points) compared to the smokers (9.0 points) (*p* = 0.021) as well as in the patients with BMI < 25 kg/m^2^ in whom the median DLQI score decreased by 18.0 points compared to the overweight/obese individuals who achieved the median DLQI reduction equal to 14.0 points (*p* = 0.023). Interestingly, in the patients with lipid disorders treated with biological agents, the median DLQI reduction from baseline was greater than in the patients without lipid disorders (median DLQI reduction from baseline 19.0 vs. 10.0, *p* = 0.012). When cyclosporine therapy was applied, a greater reduction of the median DLQI score was observed in the patients with BMI ≥ 25 kg/m^2^ in comparison to those with BMI < 25 kg/m^2^ (median DLQI reduction from baseline 9.0 vs. 5.0, *p* = 0.009).

### 3.2. PASI Score in Psoriatic Patients Prior to and after 3-Month Treatment as well as the Median PASI Reduction from Baseline Depending on the Presence or Absence of Comorbidities, BMI Score and Smoking Status

Comparison of the median PASI scores in psoriatic patients treated with different therapy regimens depending on the presence of comorbidities (hypertension, diabetes, lipid disorders), BMI score and smoking status is presented in Table 3.

Prior to launching any of the therapies, the median PASI scores at baseline were significantly different between the studied subjects with comorbidities, BMI ≥ 25 kg/m^2^ as well in smokers in comparison to the studied psoriatic patients presenting none of comorbidities mentioned above, whose BMI was below 25 kg/m^2^, also in non-smokers (*p* < 0.001). Before treatment with methotrexate, the median PASI scores were significantly higher in the patients with comorbidities and with BMI ≥ 25 kg/m^2^ than in the patients without comorbidities and with BMI < 25 kg/m^2^ (*p* < 0.001) as well as in smokers than in -non-smokers (*p* < 0.008).

In addition, before the treatment with cyclosporine was launched, the median PASI score was significantly higher in the psoriatic patients with diabetes, lipid disorders and with BMI ≥ 25 kg/m^2^ than in patients without the above named comorbidities and with BMI < 25 kg/m^2^ (*p* < 0.001, *p* = 0.004 and *p* < 0.001, respectively), whereas before treatment with retinoids, it was significantly higher in the studied patients with hypertension, diabetes, BMI ≥ 25 kg/m^2,^ and also in smokers in comparison to the patients without hypertension, diabetes, with BMI < 25 kg/m^2^, also in non-smokers (*p* = 0.006, *p* < 0.001, *p* = 0.005 and *p* < 0.001, respectively). No significant differences in the median PASI scores were noted in the patients treated with biological agents (*p* > 0.05).

After a 3-month treatment with any of the agents, significant differences were still observed in the median PASI scores between the studied subjects with comorbidities, BMI ≥ 25 kg/m^2^ or in smokers and the studied psoriatic patients presenting no comorbidities, whose BMI was below 25 kg/m^2^ as well as in non-smokers (*p* < 0.001). However, they were lower in comparison with the median PASI scores from before the treatment. As for the methotrexate therapy, after a 3-month treatment, the differences were still significant (*p* < 0.001). As for cyclosporine therapy, the presence of diabetes, lipid disorders, BMI ≥ 25 kg/m^2^ and cigarette smoking negatively affected the median PASI scores, while in the patients without diabetes, lipid disorders, with BMI < 25 kg/m^2^ and in non-smokers the median PASI scores were significantly lower (the median PASI scores 8.9 vs. 3.2, *p* < 0.001; 5.5 vs. 3.2, *p* = 0.023; 9.9 vs. 3.2, *p* < 0.001; 6.6 vs. 3.4, *p* = 0.009, respectively). In the patients treated with retinoids for 3 months, the median PASI scores were significantly higher in the studied subjects with hypertension, diabetes, BMI ≥ 25 kg/m^2^ and in the smokers (*p* < 0.008, *p* < 0.002, *p* < 0.016 and *p* < 0.002). None of the studied factors influenced the median PASI scores in the patients receiving a 3-month treatment with biological agents (*p* > 0.05).

The median PASI reduction from baseline to the control visit following a 3-month treatment was analyzed in regard to the type of therapy the studied patients received depending on the presence or absence of comorbidities (hypertension, diabetes, lipid disorders), BMI score and their smoking status. In the cyclosporine treated subgroup, the median reduction of PASI score was significantly higher in non-smokers (10.4 points) in comparison to smokers in whom it was 8.2 points (*p* = 0.045). Interestingly, in the patients treated with cyclosporine, a greater reduction of the median PASI score was observed in the patients with lipid disorders and with BMI ≥ 25 kg/m^2^ than in those without lipid disorders, with BMI < 25 kg/m^2^ and in non-smokers (*p* = 0.023, *p* = 0.043 and *p* = 0.045, respectively). When treatment with methotrexate was applied, the psoriatic patients also suffering from hypertension achieved greater median PASI score reduction from baseline than the patients who did not have this comorbidity (*p* = 0.025).

### 3.3. Assessment of PASI Score Improvement Expressed by Achieving PASI50, PASI75 and PASI90 by Psoriatic Patientstreated with Different Therapy Regimens Depending on the Presence or Absence of Comorbidities, BMI Score and Smoking Status

The number and percentages of psoriatic patients treated with methotrexate, cyclosporine, retinoids and biological agents achieving PASI50, PASI75 and PASI90 depending on the presence or absence of comorbidities, BMI score and smoking status are presented in Table 4.

PASI90 was achieved independently of the presence or absence of the analyzed comorbidities, BMI score and smoking status in all the studied patients as well as in those undergoing individual therapies, i.e., therapies with methotrexate, cyclosporine, retinoids or biological agents. However, achieving PASI75 and PASI50 depended on the presence or absence of the analyzed factors. In the whole studied group, the number and percentages of patients achieving PASI75 were significantly higher in the absence of hypertension, diabetes, lipid disorders, also in those patients whose BMI was below 25 kg/m^2^, as well as in the non-smokers in comparison to the patients who had hypertension, diabetes, lipid disorders, BMI ≥ 25 kg/m^2^ as well as in the smoking patients (*p* < 0.001, *p* < 0.001, *p* = 0.006, *p* = 0.004 and *p* < 0.001, respectively). Moreover, in the patients treated with methotrexate, the presence of analyzed comorbidities, BMI ≥ 25 kg/m^2^ and cigarette smoking had a significant negative effect on the number and percentages of patients achieving PASI75. The presence of diabetes, BMI ≥ 25 kg/m^2^ and smoking had a negative effect on achieving PASI75 in the patients treated with cyclosporine (*p* < 0.001, *p* = 0.013 and *p* = 0.016, respectively).

The number and percentages of patients achieving PASI50 were significantly higher in the absence of hypertension, diabetes, also when BMI < 25 kg/m^2^ as well as in the non-smokers in comparison to the patients with hypertension, diabetes, BMI ≥ 25 kg/m^2^, and in those studied subjects who smoked cigarettes (*p* < 0.001). In the patients treated with methotrexate, the presence of analyzed comorbidities, BMI ≥ 25 kg/m^2^ and cigarette smoking had a significant negative effect on the number and percentages of patients achieving PASI50. Moreover, in the cyclosporine treated patients, a negative effect on achieving PASI50 was observed in the presence of diabetes, when BMI was equal or above 25 kg/m^2^ and in the smoking patients.

No significant differences in achieving PASI50, PASI75 and PASI90 depending on the presence or absence of comorbidities, BMI score and smoking status were found with regard to the number and percentages of psoriatic patients treated with retinoids and biological agents.

## 4. Discussion

Psoriasis is a multisystem autoinflammatory disease which affects not only the physical health, but it also has an extensive emotional and psychosocial effect on psoriatic patients. Since hypertension, diabetes, lipid disorders as well as overweight/obesity and nicotine dependence are more prevalent in psoriatic patients than in the general population, they were selected as representative comorbidities to investigate their impact on health-related quality of life (HRQOL) and psoriasis severity in psoriatic patients receiving systemic conventional treatment with methotrexate, cyclosporine, retinoids and biological treatment. In order to determine the efficacy of psoriasis treatment in the presence of the aforementioned comorbidities, the DLQI and PASI scores were assessed prior to launching a selected therapy and after 3 months of treatment.

Prior to the treatment, the median DLQI and PASI scores were significantly higher in the study subjects with comorbidities (hypertension, diabetes, lipid disorders), the scores were also significantly higher in the patients with BMI ≥ 25 kg/m^2^ as well as in the smokers in comparison with the study subjects who had no comorbidities, whose BMI was below 25 kg/m^2^, and also in the non-smokers, which is reflective of the negative effect of the co-existing diseases, overweight/obesity and smoking on HRQOL and psoriasis severity. Analysis of all types of applied medical treatment revealed that after a 3-month treatment, the median DLQI and PASI scores were still significantly higher in the patients with comorbidities, BMI ≥ 25 kg/m^2^, and also in the smoking patients in comparison to those who did not have hypertension, diabetes or lipid disorders, and whose BMI was below 25 kg/m^2^ as well as in the non-smokers. After a 3-month treatment, similar data were observed for the patients treated with methotrexate, however, the median DLQI score in the patients treated with cyclosporine was significantly higher only in those who had diabetes, or were overweight/obese whereas in the patients treated with retinoids only when they had hypertension, diabetes or smoked cigarettes.

Michalska-Bańkowska et al., observed that concommitant diabetes and a metabolic syndrome did not affect the efficacy of cyclosporine in psoriasis treatment. However, in the study they included thirty-two patients with psoriasis vulgaris, of which seven had coexisting diabetes mellitus, seven patients had both diabetes and metabolic syndrome, and the remaining patients had no comorbidities [22].

As for the analysis of the median PASI score after a 3-month treatment, it was significantly higher in those studied psoriatic patients treated with cyclosporine who also had diabetes, lipid disorders or who were overweight/obese as well as in those who were smokers. The median PASI score after a 3-month treatment was significantly higher in those psoriatic patients treated with retinoids who also suffered from hypertension, diabetes or who were overweight/obese, also in those who were smokers. In the study of Hayran Y. and Yalcin B., 67.6% of the psoriasis patients were smokers. The authors observed that smoking had a negative effect on psoriasis severity expressed by PASI. Among smokers, 29.3% of the patients had moderate/severe psoriasis (*p* = 0.028). The median PASI score was 5.9 for smokers compared to 2.3 in non-smokers (*p* = 0.003) [23]

All the conventional treatment outcomes, i.e., methotrexate, cyclosporine and retinoids therapies, are indicative of the negative impact of the analyzed factors, i.e., comorbidities, overweight/obesity, nicotine dependence, on the DLQI and PASI scores both before launching the treatment and after a 3-month treatment. The DLQI score in the study of Sanchez-Carazo confirmed that patients with several comorbidities, such as obesity and arterial hypertension, had worse scores in the physical component of the quality-of-life questionnaire [24]. Interestingly, our study showed that the outcome of therapy with biological agents was independent of each of the analyzed factors.

Having analyzed the DLQI and PASI scores prior to and after a 3-month treatment with conventional and biological agents, the median DLQI and PASI reduction from baseline was calculated. It was revealed that the medium PASI reduction from baseline in the patients with hypertension and treated with methotrexate was significantly higher than in those patients who did not have hypertension. However, the median PASI score prior to the treatment was nearly twice as high in comparison to the patients without hypertension. The result may prove that treatment with methotrexate is effective even if patients had co-existing hypertension. Moreover, the patients with overweight/obesity treated with cyclosporine presented a greater reduction in the DLQI and PASI scores than the patients with BMI < 25 kg/m^2^. Interestingly, a greater reduction of PASI was also observed in the patients treated with cyclosporine who also had lipid disorders as well as in those patients who did not smoke in comparison to the patients who had no lipid disorders and in smokers. What is more, significant differences in the median DLQI reduction were observed in the patients treated with biological agents who had lipid disorders but who were not overweight/obese and did not smoke in comparison to the patients without lipid disorders, with BMI < 25kg/m^2^ and who were smokers. No differences in the median PASI reduction were observed in the patients treated with biological agents. These findings suggest that in the patients treated with biological agents the skin lesions’ improvement, but not quality of life, was independent of the presence or absence of comorbidities, BMI score and smoking status. However, these findings may be distorted by a small number of the studied psoriatic patients treated with biological agents who also suffered from hypertension (10%), had lipid disorders (10%), who were smokers (25%), and whose BMI was below 25 kg/m^2^ (10%). In the study of Prignano et al., all biological treatments significantly reduced PASI score. The authors included 471 patients with plaque psoriasis and psoriatic arthritis. In both psoriasis groups, the main comorbidity was hypertension, while hyperlipidaemia was frequently present in the patients treated with biologics [25].

Analysis of all the studied psoriatic patients revealed that the number and percentages of the patients achieving PASI75 and PASI50 were significantly higher when the patients did not suffer from hypertension, diabetes, lipid disorders, when their BMI was below 25 kg/m^2^ as well as in those who did not smoke. Similar significant differences were observed in the psoriatic patients treated with methotrexate. However, in the patients treated with cyclosporine, the number and percentages of patients achieving PASI75 and PASI50 were significantly different and had a negative impact only in the patients with diabetes, overweight/obesity and in the smokers. Interestingly, in the psoriatic patients treated with either retinoids or biological agents, no significant differences in achieving PASI75 and PASI50 were found in the patients both with and without the analyzed factors. Petridis et al., in a multicentre, prospective, observational study which included 136 psoriatic patients treated with infliximab found out that the BMI score did not have any impact on the PASI75 or DLQI improvement rate [26]. Anzengruber et al., in the group of 841 smokers and 423 non-smokers included in national psoriasis registries for Germany and Switzerland, found that smoking did not affect the therapy response of psoriatic skin lesions to systemic anti-psoriatic therapies after 3, 6 and 12 months [27]. Warren et al., in a multicentre longitudinal cohort study, identified eight factors, among others, ex- and current smoking, high weight, which were negatively associated with achieving ≥90% improvement in PASI at six months of treatment [28]. As for the results of our study, achieving PASI90 was not significantly different in any of the therapies regardless of the patients the presence or absence of the analyzed factors.

## 5. Conclusions

Psoriatic patients with comorbidities such as hypertension, diabetes, lipid disorders, overweight/obesity as well as those study subjects who were smokers had poorer HRQOL and presented more severe psoriatic skin lesions. The presence of comorbidities, overweight/obesity and nicotine dependence had a negative effect on the outcome of conventional treatment with methotrexate, cyclosporine, retinoids. The outcome of therapy with biological agents, however, was independent of each of the analyzed factors. Biological agents were highly effective in the treatment of psoriatic skin lesions despite the presence of comorbidities, whereas methotrexate was effective even if patients had co-existing hypertension.

*Limitations*: A limitation to the study may be a different number of patients treated with conventional therapies, i.e., methotrexate, cyclosporine, retinoids, and biological agents. Most patients were treated with methotrexate (*n* = 79), and the number of patients treated with biological agents was much smaller (*n* = 20). The each of study group differed with regard to their sex, age, duration of psoriasis and different dosages of treatment. The studied patients came from different residential areas, had different marital, professional status and other previous treatment. The number and percentage of patients with comorbidities, overweight/obesity, and nicotine dependence varied among patients treated with methotrexate, cyclosporine, retinoids and biological agents.

## Figures and Tables

**Table 1 ijerph-18-13167-t001:** Characteristics of the studied psoriatic patients before the treatment with the use of different applied therapies.

Characteristics	Variable	All Applied Therapies (*n* = 184)	Methotrexate (*n* = 79)	Cyclosporine (*n* = 42)	Retinoids (*n* = 43)	Biological Agents (*n* = 20)
Gender	Male, *n* (%)	97 (52.7%)	47 (59.4%)	10 (23.8%)	32 (74.4%)	8 (40%)
	Female, *n* (%)	87 (47.3%)	32 (40.6%)	32 (76.2%)	11 (25.6%)	12 (60%)
Age (years)	Mean ± SD	46 ± 12	47 ± 10	43 ± 10	47 ± 10	39 ± 3
	range	(20–80)	(20–68)	(29–62)	(25–68)	(34–43)
	18–40, *n* (%)	76 (41.3%)	26 (32.9%)	22 (52.3%)	14 (32.6%)	14 (70%)
	41–60, *n* (%)	95 (51.6%)	44 (55.7%)	18 (42.9%)	27 (62.8%)	6 (30%)
	≥61, *n* (%)	13 (7.1%)	9 (11.4%)	2 (4.8%)	2 (4.6%)	0
Place of residence	City, *n* (%)	110 (59.8%)	48 (60.8%)	24 (57.1%)	27 (62.8%)	11 (55%)
	Village, *n* (%)	74 (40.2%)	31 (39.2%)	18 (42.9%)	16 (37.2%)	9 (45%)
Marital status	Single, *n* (%)	56 (30.4%)	22 (27.8%)	14 (33.3%)	13 (30.2%)	7 (35%)
	Married, *n* (%)	110 (59.8%)	49 (62%)	26 (61.9%)	22 (51.2%)	13 (65%)
	Widowed, *n* (%)	18 (9.8%)	8 (10.2%)	2 (4.8%)	8 (18.6%)	0
Employment	Employed, *n* (%)	115 (62.5%)	49 (62%)	23 (54.8%)	28 (65.1%)	15 (75%)
	Unemployed, *n* (%)	41 (22.3%)	14 (17.7%)	12 (28.6%)	10 (23.3%)	5 (25%)
	Pensioner, *n* (%)	28 (15.2%)	16 (20.3%)	7 (16.6%)	5 (11.6%)	0
Smoking status	Smoker, *n* (%)	65 (35.3%)	35 (44.3%)	11 (26.2%)	14 (32.6%)	5 (25%)
	Non-smoker, *n* (%)	119 (64.7%)	44 (55.7%)	31 (73.8%)	29 (67.4%)	15 (75%)
BMI (kg/m^2^)	Median (IQR)	25.0 (23–29)	25.3 (23–30)	23.0 (22–27)	24.0 (23–25.7)	27.0 (25.5–29.5)
	≥25, *n* (%)	92 (50%)	44 (55.7%)	12 (38.6%)	18 (41.9%)	18 (90%)
	<25, *n* (%)	92 (50%)	35 (44.3%)	30 (71.4%)	25 (58.1%)	2 (10%)
Disease duration (years)	Mean ± SD	15 ± 10	14 ± 9	12 ± 8	18 ± 13	11 ± 4
Presence of comorbidities	Hypertension, *n* (%)	62 (33.7%)	37 (46.8%)	5 (11.9%)	18 (41.9%)	2 (10%)
	Diabetes, *n* (%)	40 (21.7%)	22 (27.8%)	12 (28.6%)	6 (14%)	0
	Lipid disorders, *n* (%)	60 (32.6%)	38 (48.1%)	20 (47.6%)	0	2 (10%)
DLQI	Median (IQR)	11.0 (7–20)	10.0 (6–20)	13.0 (8–20)	15.0 (10–21)	20.0 (18–24)
	<10, *n* (%)	48 (26.1%)	29 (36.1%)	11 (26.2%)	8 (18.6%)	0 (0%)
	≥10, *n* (%)	136 (73.9%)	50 (63.9%)	31 (73.8%)	35 (81.4%)	20 (100%)
PASI	Median (IQR)	14.6 (12.2–22.6)	18.4 (14.7–28)	14.2 (12.6–18.2)	13.8 (12.2–15.7)	27.3 (23–30)
	<10, *n* (%)	1 (0.5%)	0 (0%)	0 (0%)	1 (2.3%)	0 (0%)
	≥10, *n* (%)	183 (99.5%)	79 (100%)	42 (100%)	42 (97.7%)	20 (100%)
BSA	Median (IQR)	25.0 (20–32)	29.0 (22–37)	25.0 (22–28)	25.0 (21–28)	38.0 (32–39)
	<10, *n* (%)	0 (0%)	0 (0%)	0 (0%)	0 (0%)	0 (0%)
	≥10, *n* (%)	184 (100%)	79 (100%)	42 (100%)	43 (100%)	20 (100%)

**Table 2 ijerph-18-13167-t002:** The DLQI score before the treatment (a baseline visit), after a 3-month therapy (a control visit) as well as DLQI reduction in psoriasis patients treated with different therapies depending on the presence of comorbidities, BMI score and smoking status.

DLQI Score Before the Treatment (a Baseline Visit)						
Analyzed Factor, Me (Min-Max)		All Appled Therapies (*n* = 184)	Methotrexate (*n* = 79)	Cyclosporine (*n* = 42)	Retinoids (*n* = 43)	Biological Agents (*n* = 20)
Hypertension	Yes (*n*-62)	15 (9–21)	18 (10–20)	21 (15–24)	19 (11–24)	22 (-)
	No (*n*-122)	10 (6–18)	7 (5–10)	12 (8–20)	14 (10–18)	20 (18–24)
*p*		0.004	<0.001	0.213	0.089	0.612
Diabetes	Yes (*n*-40)	19 (10–21)	20 (11–21)	20 (15–22)	24 (22–24)	-
	No (*n*-144)	10 (6–18)	10 (5–14)	10 (6–20)	14 (10–19)	-
*p*		0.012	<0.001	0.014	0.027	-
Lipid disorders	Yes (*n*-60)	17 (10–20)	18 (10–21)	18 (10–21)	-	21.0 (-)
	No (*n*-124)	10 (6–20)	7 (5–10)	11 (6–20)	-	20.0 (18–24)
*p*		0.014	<0.001	0.307	-	0.969
BMI (kg/m^2^)	≥25 (*n*-92)	17 (9–20)	14 (10–21)	18 (18–21)	17 (10–24)	25.0 (24–26)
	<25 (*n*-92)	10 (6–17)	8 (5–10)	10 (6–20)	14 (10–20)	20.0 (18–22)
*p*		<0.001	<0.001	0.053	0.330	0.080
Smoking status	Smoker (*n*-65)	16 (10–20)	14 (10–20)	18 (12–22)	20 (15–26)	18.0 (16–22)
	Non-smoker (*n*-119)	10 (6–20)	10 (5–16)	11 (7–20)	12 (10–18)	22.0 (20–24)
*p*		0.006	0.008	0.169	0.007	0.148
**DLQI Score after a 3-Month Therapy (a Control Visit)**						
**Analyzed factor, Me (Min-Max)**		**All Appled Therapies** **(*n* = 184)**	**Methotrexate** **(*n* = 79)**	**Cyclosporine** **(*n* = 42)**	**Retinoids** **(*n* = 43)**	**Biological Agents** **(*n* = 20)**
Hypertension	Yes (*n*-62)	10 (7–14)	10 (7–14)	10 (8–15)	12 (6–15)	7 (-)
	No (*n*-122)	6 (4–8)	4 (2–6)	6 (5–10)	6 (5–9)	8 (6–9)
*p*		<0.001	<0.001	0.150	0.013	0.658
Diabetes	Yes (*n*-40)	10 (7–15)	12 (7–18)	10 (8–13)	15 (13–19)	-
	No (*n*-144)	6 (4–9)	5 (3–8)	6 (4–7)	7 (5–11)	-
*p*		<0.001	<0.001	0.003	0.013	-
Lipid disorders	Yes (*n*-60)	10 (6–12)	10.0 (6–15)	8 (6–12)	-	7 (-)
	No (*n*-124)	6 (4–9)	4.0 (3–7)	6 (4–8)	-	8 (6–9)
*p*		<0.001	<0.001	0.150	-	0.658
BMI (kg/m^2^)	≥25 (*n*-92)	8 (6–11)	9 (5–14)	10 (7–11)	8 (6–15)	8 (6–9)
	<25 (*n*-92)	6 (4–8)	4 (2–7)	6 (4–8)	8 (6–12)	7 (7–8)
*p*		<0.001	<0.001	0.021	0.422	0.658
Smoking status	Smoker (*n*-65)	9 (6–14)	10 (5–14.5)	9 (6.5–14)	13 (6–19)	9 (8–10)
	Non-smoker (*n*-119)	6 (4–9)	4 (3–8)	6 (4–9)	7 (5–10)	7 (6–9)
*p*		<0.001	<0.001	0.088	0.017	0.073
**DLQI Reduction from Baseline**						
**Analyzed factor, Me (Min-Max)**		**All Appled Therapies** **(*n* = 184)**	**Methotrexate** **(*n* = 79)**	**Cyclosporine** **(*n* = 42)**	**Retinoids** **(*n* = 43)**	**Biological Agents** **(*n* = 20)**
Hypertension	Yes (*n*-62)	4 (1–7)	4 (2–7)	8 (6–11)	6 (4–8)	15 (-)
	No (*n*-122)	5 (2–9)	3 (1–6)	6 (4–7)	6 (4–8)	14 (11–15)
*p*		0.064	0.280	0.403	0.758	0.255
Diabetes	Yes (*n*-40)	5 (2–7)	5 (2–7)	8 (6–11)	8 (5–9)	-
	No (*n*-144)	4 (1–8)	3 (1–6)	5 (3–9)	6 (4–8)	-
*p*		0.690	0.340	0.094	0.334	-
Lipid disorders	Yes *n*-(60)	4 (2–8)	5 (2–7)	7 (4–10)	-	19 (10–21)
	No (*n*-124)	5 (1–8)	3 (1–6)	5 (4–11)	-	10 (6–18)
*p*		0.657	0.104	0.545	-	0.012
BMI (kg/m^2^)	≥25 (*n*-92)	5 (1–9)	4 (1–7)	9 (7–12)	7 (5–9)	14 (11–15)
	<25 (*n*-92)	4 (2–7)	4 (2–7)	5 (3–7)	5 (4–8)	18 (17–18)
*p*		0.364	0.988	0.009	0.408	0.023
Smoking status	Smoker (*n*-65)	5 (1–8)	4 (1–6)	8 (5–9)	8 (5–9)	9 (8–12)
	Non-smoker (*n*-119)	5 (3–10)	4 (2–7)	5 (3–10)	6 (3–8)	14 (13–15)
*p*		0.126	0.491	0.322	0.104	0.021

**Table 3 ijerph-18-13167-t003:** The PASI score before the treatment (a baseline visit), after a 3-month therapy (a control visit) as well as PASI reduction in psoriasis patients treated with different therapies depending on the presence of comorbidities, BMI score and smoking status therapies depending on the presence of comorbidities, BMI score and smoking status.

PASI Score before the Treatment (a Baseline Visit)						
Analyzed factor, Me (Min-Max)		All appled therapies (*n* = 184)	Methotrexate (*n* = 79)	Cyclosporine (*n* = 42)	Retinoids (*n* = 43)	Biological Agents (*n* = 20)
Hypertension	Yes (*n*-62)	18.4 (13.8–27)	27.4 (22.3–34.6)	21.4 (14–22.4)	14.9 (13.8–21.2)	25.2 (-)
	No (*n*-122)	14.2 (11.2–18.6)	15.2 (13.6–18)	14.2 (12.6–16.6)	12.8 (10.8–14.2)	28.1 (22.4–29.7)
*p*		<0.001	<0.001	0.130	0.006	0.614
Diabetes	Yes (*n*-40)	22.1 (14.4–28.6)	30.6 (27–36.2)	21.5 (16.8–24)	21.3 (17.2–23.4)	-
	No (*n*-144)	14.2 (11.2–18.8)	16.8 (14.2–21.3)	13.8 (12.2–14.4)	13.4 (11.1–14.5)	-
*p*		<0.001	<0.001	<0.001	<0.001	-
Lipid disorders	Yes *n*-(60)	20.4 (14.1–27.7)	27.5 (20.8–35)	16.7 (14–22)	-	26.0 (-)
	No (*n*-124)	14.2 (11–17.9)	16.0 (14–17.9)	13.7 (11.8–17.2)	-	28.1 (22.4–29.7)
*p*		<0.001	<0.001	0.004	-	0.753
BMI (kg/m^2^)	≥25 (*n*-92)	20.8 (14–27)	26.7 (18.4–31.6)	21.5 (17.3–24)	15.0 (12.8–21.2)	27.6 (22.4–29.7)
	<25 (*n*-92)	13.8 (11–15)	15.0 (13.7–17.7)	13.7 (12.2–14.4)	13.2 (10.8–14.2)	25.8 (23–27)
*p*		<0.001	<0.001	<0.001	0.005	0.706
Smoking status	Smoker (*n*-65)	19.3 (13.8–26.7)	26.4 (18.4–33.8)	16.4 (13.5–21.5)	16.8 (14.2–21.4)	28.4 (23.4–31.1)
	Non-smoker (*n*-119)	14.2 (10.8–18.4)	16.4 (14.3–25.3)	14.2 (12.5–16.1)	12.8 (10.8–14.2)	26.8 (23.4–29.6)
*p*		<0.001	<0.001	0.208	<0.001	0.827
**PASI Score after a 3-Month Therapy (a control Visit)**						
**Analyzed factor, Me (Min-Max)**		**All Appled Therapies** **(*n* = 184)**	**Methotrexate** **(*n* = 79)**	**Cyclosporine** **(*n* = 42)**	**Retinoids** **(*n* = 43)**	**Biological Agents** **(*n* = 20)**
Hypertension	Yes (*n*-62)	10.0 (6.9–15.2)	13.4 (9.2–19.4)	10.8 (3.5–14.2)	7.9 (6.8–11.2)	5.8 (-)
	No (*n*-122)	4.8 (3.2–6.6)	4.2 (3.2–6)	4.4 (2.3–6.3)	5.8 (3.8–7.9)	6.2 (4.7–8.2)
*p*		<0.001	<0.001	0.095	0.008	0.614
Diabetes	Yes (*n*-40)	11.2 (7.4–16)	16.0 (13.4–23)	8.9 (6.3–11.3)	13.2 (10.6–16.6)	-
	No (*n*-144)	5.2 (3.4–7.2)	4.8 (3.4–8.4)	3.2 (1.6–5.2)	6.4 (4.7–8.2)	-
*p*		<0.001	<0.001	<0.001	0.002	-
Lipid disorders	Yes (*n*-60)	10.0 (5.1–15)	14.3 (9–20)	5.5 (3.6–9.9)	-	5.8 (-)
	No (*n*-124)	5.2 (3.4–7.4)	4.4 (3.4–6.1)	3.2 (1.6–5.6)	-	6.2 (4.7–8.2)
*p*		<0.001	<0.001	0.023	-	0.706
BMI (kg/m^2^)	≥25 (*n*-92)	8.2 (5.8–13.4)	11.6 (7.3–19.6)	9.9 (6.6–11.3)	7.9 (6.4–15.2)	6.4 (5.2–8.2)
	<25 (*n*-92)	4.4 (3.2–6.2)	4.2 (3.3–5.7)	3.2 (1.6–5.2)	6.2 (3.7–7.9)	4.1 (2.4–5.8)
*p*		<0.001	<0.001	<0.001	0.016	0.208
Smoking status	Smoker (*n*-65)	9.8 (6.3–15)	13.2 (7–19)	6.6 (5.3–11.1)	9.7 (7–16.6)	6.5 (5.2–8.5)
	Non-smoker (*n*-119)	4.8 (3.2–7.2)	4.2 (3.4–9)	3.4 (1.95–5.65)	6.2 (3.8–7.7)	5.8 (4.8–7.9)
*p*		<0.001	<0.001	0.009	0.002	0.600
**PASI Reduction from Baseline**						
**Analyzed factor, Me (Min-Max)**		**All Appled Therapies** **(*n* = 184)**	**Methotrexate** **(*n* = 79)**	**Cyclosporine** **(*n* = 42)**	**Retinoids** **(*n* = 43)**	**Biological Agents** **(*n* = 20)**
Hypertension	Yes (*n*-62)	8.4 (4–12.4)	12.4 (11–15.2)	9.8 (8.2–10.7)	6.1 (4.8–8.2)	19.5 (-)
	No (*n*-122)	9.8 (7–13)	11.1 (9–13)	10.4 (9.2–11.4)	7.0 (5.2–8.2)	20.8 (17.7–22.3)
*p*		0.079	0.025	0.268	0.546	0.674
Diabetes	Yes (*n*-40)	10.3 (4.8–12)	12.0 (11–15.6)	10.9 (9.3–12.8)	6.8 (4.8–8.2)	-
	No (*n*-144)	9.4 (6–13)	11.6 (9.3–13.2)	9.9 (9.2–11.2)	6.4 (5.2–8.2)	-
*p*		0.518	0.255	0.277	0.972	-
Lipid disorders	Yes (*n*-60)	11.0 (7.3–13)	12.1 (11–15)	10.8 (10.1–11.4)	-	20.2 (-)
	No (*n*-124)	8.6 (5.4–12.1)	11.2 (9–13.5)	9.3 (8.2–11)	-	20.1 (17.7–22.2)
*p*		0.053	0.182	0.023	-	0.974
BMI (kg/m^2^)	≥25 (*n*-92)	10.6 (4.9–14.1)	11.6 (10.5–13.6)	11.0 (10.5–12.8)	6.4 (5.6–7.8)	19.9 (17.7–22.3)
	<25 (*n*-92)	9.2 (6.4–11.4)	11.6 (9–13.5)	9.8 (9.2–11)	6.8 (4.8–8.3)	21.7 (21.4–22)
*p*		0.206	0.629	0.043	0.854	0.516
Smoking status	Smoker (*n*-65)	8.4 (4.8–12.1)	11.2 (9–13.2)	8.2 (8.2–10.8)	6.0 (4.8–7.8)	20.2 (15.9–22.4)
	Non-smoker (*n*-119)	10.0 (6–13)	12.0 (10.3–13.9)	10.4 (9.4–11.4)	6.8 (5.4–8.3)	20.0 (18.2–22.2)
*p*		0.285	0.356	0.045	0.476	0.800

**Table 4 ijerph-18-13167-t004:** The number and percentage of psoriatic patients achieving 90%, 75% and 50% improvement in PASI score after a 3-month treatment with different therapies depending on the presence of comorbidities, BMI score and smoking status on the presence of comorbidities, BMI score and smoking status.

Analyzed Factor		PASI90, *n* (%)	PASI75, *n* (%)	PASI50, *n* (%)
All Applied Therapies				
Hypertension	Yes	0/62	5/62 (8.1%)	34/62 (54.8%)
	No	4/122 (3.3%)	50/122 (41%)	114/122 (93.4%)
*p*		0.302	<0.001	<0.001
Diabetes	Yes	0/40	0/40	20/40 (50%)
	No	4/144 (2.8%)	55/144 (38.2%)	128/144 (88.9%)
*p*		0.578	<0.001	<0.001
Lipid disorders	Yes	1/60 (1.7%)	10/60 (16.7%)	44/60 (73.3%)
	No	3/124 (2.4%)	45/124 (36.3%)	104/124 (83.9%)
*p*		ns	0.006	0.113
BMI (kg/m^2^)	≥25	0/92	18/92 (19.6%)	65/92 (70.7%)
	<25	4/92 (4.3%)	37/92 (40.2%)	83/92 (90.2%)
*p*		0.121	0.004	<0.001
Smoking status	Smoker	1/65 (1.5%)	4/65 (6.2%)	39/65 (60%)
	Non-smoker	3/119 (2.5%)	51/119 (42.9%)	109/119 (91.6%)
*p*		ns	<0.001	<0.001
**Methotrexate**				
Hypertension	Yes	0/37	1/37 (2.7%)	23/37 (62.2%)
	No	0/42	17/42 (40.5%)	39/42 (92.9%)
*p*		ns	<0.001	0.002
Diabetes	Yes	0/22	0/22	9/22 (40.9%)
	No	0/57	18/57 (31.6%)	53/57 (93%)
*p*		ns	0.002	<0.001
Lipid disorders	Yes	0/38	2/38 (5.3%)	22/38 (57.9%)
	No	0/41	16/41 (39%)	40/41 (97.6%)
*p*		ns	<0.001	<0.001
BMI (kg/m^2^)	≥25	0/44	4/44 (9.1%)	27/44 (61.4%)
	<25	0/35	14/35 (40%)	35/35 (100%)
*p*		ns	0.002	<0.001
Smoking status	Smoker	0/35	1/35 (2.9%)	22/35 (62.9%)
	Non-smoker	0/44	17/44 (38.6%)	40/44 (90.9%)
*p*		ns	<0.001	0.005
**Cyclosporine**				
Hypertension	Yes	0/5	1/5 (20%)	5/5 (100%)
	No	4/37 (10.8%)	16/37 (43.2%)	34/37 (91.9%)
*p*		ns	0.632	ns
Diabetes	Yes	0/12	0/12	9/12 (75%)
	No	4/30 (13.3%)	17/30 (56.7%)	30/30 (100%)
*p*		0.308	<0.001	0.019
Lipid disorders	Yes	1/20 (5%)	6/20 (30%)	18/20 (90%)
	No	3/22 (13.6%)	11/22 (50%)	21/22 (95.5%)
*p*		0.608	0.222	0.598
BMI (kg/m^2^)	≥25	0/12	1/12 (8.3%)	9/12 (75%)
	<25	4/30 (13.3%)	16/30 (53.3%)	30/30 (100%)
*p*		0.308	0.013	0.019
Smoking status	Smoker	1/11 (9.1%)	1/11 (9.1%)	8/11 (72.7%)
	Non-smoker	3/31 (9.7%)	16/31 (51.6%)	31/31 (100%)
*p*		ns	0.016	0.014
**Retinoids**				
Hypertension	Yes	0/18	1/18 (5.6%)	7/18 (38.9%)
	No	0/25	5/25 (20%)	15/25 (60%)
*p*		ns	0.375	0.223
Diabetes	Yes	0/6	0/6	2/6 (33.3%)
	No	0/37	6/37 (16.2%)	20/37 (54.1%)
*p*		ns	0.571	0.413
Lipid disorders	Yes	-	-	-
	No	-	-	-
*p*		-	-	-
BMI (kg/m^2^)	≥25	0/18	1/18 (5.6%)	9/18 (50%)
	<25	0/25	5/25 (20%)	13/25 (52%)
*p*		ns	0.375	ns
Smoking status	Smoker	0/14	0/14	4/14 (28.6%)
	Non-smoker	0/29	6/29 (20.7%)	18/29 (62.1%)
*p*		ns	0.155	0.055
**Biological Agents**				
Hypertension	Yes	0/2	2/2 (100%)	2/2 (100%)
	No	0/18	12/18 (66.7%)	18/18 (100%)
*p*		ns	ns	ns
Diabetes	Yes	-	-	-
	No	-	-	-
*p*		-	-	-
Lipid disorders	Yes	0/2	2/2 (100%)	2/2
	No	0/18	12/18 (66.7%)	18/18
*p*		ns	ns	ns
BMI (kg/m^2^)	≥25	0/18	12/18 (66.7%)	18/18 (100%)
	<25	0/2	2/2 (100%)	2/2 (100%)
*p*		ns	ns	ns
Smoking status	Smoker	0/5	2/5 (40%)	5/5
	Non-smoker	0/15	12/15 (80%)	15/15
*p*		ns	0.131	ns

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
