# Peer review of "The Impact of Hypertension, Diabetes, Lipid Disorders, Overweight/Obesity and Nicotine Dependence on Health-Related Quality of Life and Psoriasis Severity in Psoriatic Patients Receiving Systemic Conventional and Biological Treatment"

_ijerph, 2021, doi:10.3390/ijerph182413167_

Round 1
Reviewer 1 Report
The study was conducted between 2019 and 2020. It included 184 patients with plaque psoriasis (97 males and 87 females). In addition, information regarding health-related quality of life (HRQOL), Psoriasis Area Severity Index (PASI), Body Surface Area (BSA), Dermatology Life Quality Index (DLQI), hypertension, diabetes, lipid disorders, BMI, and smoking status were all collected. This study attempted to investigate the effect of comorbidities on HRQOL and psoriasis severity measured appropriately by the DLQI and PASI before, after a 3-month treatment and the median DLQI or PASI reduction from baseline in the adult psoriatic patients receiving various types of treatment. Authors concluded that patients with comorbidities have poorer HRQOL and more severe skin damage in psoriasis. The presence of comorbidities has a negative impact on the outcome of treatment with conventional therapies. I totally agreed that this study could provide some information for the clinicians. Nevertheless, I still have a few major and minor concerns.
Major concerns
Q1. According to the table in the manuscript, authors have displayed some demographic characteristics, such as age, gender, and place of residence. However, this study only carried out the non-parametric tests for relevant analyses. Authors should consider these factors in the regression models to avoid the potential confounding effects.
Q2. Some factors were not considered, such as economic status and alcohol use, which were also associated with the effectiveness of psoriasis treatment.
Q3. The definitions of PASI and BSA should be clearly explained in the manuscript so that readers from different fields could understand the research methods.
Q4. Authors decided to compare the psoriasis severity before, after a ‘3-month treatment’. It was still unclear why authors used this treatment interval. Authors should provide clear information for the authors.
Minor concerns
Q5. In Materials and Methods, using the percentage to present the data should be '.' instead of ','.
Q6. In Discussion, it was inappropriate to repeat the detailed results of this study. Authors should review more previous studies for relevant issues and provide the clear statements in Discussion.
Author Response
The study was conducted between 2019 and 2020. It included 184 patients with plaque psoriasis (97 males and 87 females). In addition, information regarding health-related quality of life (HRQOL), Psoriasis Area Severity Index (PASI), Body Surface Area (BSA), Dermatology Life Quality Index (DLQI), hypertension, diabetes, lipid disorders, BMI, and smoking status were all collected. This study attempted to investigate the effect of comorbidities on HRQOL and psoriasis severity measured appropriately by the DLQI and PASI before, after a 3-month treatment and the median DLQI or PASI reduction from baseline in the adult psoriatic patients receiving various types of treatment. Authors concluded that patients with comorbidities have poorer HRQOL and more severe skin damage in psoriasis. The presence of comorbidities has a negative impact on the outcome of treatment with conventional therapies. I totally agreed that this study could provide some information for the clinicians. Nevertheless, I still have a few major and minor concerns.
Answer to the Reviewer: Thank you for your evaluation. I would like to answer for your questions.
Major concerns
Q1. According to the table in the manuscript, authors have displayed some demographic characteristics, such as age, gender, and place of residence. However, this study only carried out the non-parametric tests for relevant analyses. Authors should consider these factors in the regression models to avoid the potential confounding effects.
Answer to the Reviewer: Thank you for the valuable comment. We fully agree with the reviewer's
opinion that data presented in the table 1 can suggest that numerous clinical-demographic data are able to alter the results presented in the next tables of the paper. Nevertheless, prior to analysis we checked whether aforementioned factors affect the either DLQI or PASI related
results. For this purpose the multiple regression model including clinical-demographic features of studied individuals was constructed. None of the parameters has impact of the test result (all p over the 0.05). To be more clear we added the following sentences to material and methods
section:
Statistical analysis was conducted with the use of the MedCalc computer software, version 18.2.1 (MedCalc, Belgium). Data distribution in the studied population was examined by the Shapiro-Wilk test. All the studied parameters demonstrated abnormal distribution of the data (p<0.05)
therefore, non-parametric tests were applied in the further analysis. Multiple regression analysis for the clinical-demographic features of the studied individuals was applied to avoid the potential confounding effects on the results. None of the demographic factors affected either DLQI or
PASI score, thus the whole studied group underwent non-parametric testing. Values of the studied factors in the subgroups of patients treated with different therapy regimens were expressed by the median (Me) and interquartile range (IQR). Differences in the values of the studied parameters among the patients in regard to the presence of comorbidities, BMI score and smoking status were tested by the Mann-Whitney U test. The results presenting p values below 0.05 were considered as statistically significant.
Q2. Some factors were not considered, such as economic status and alcohol use, which were also associated with the effectiveness of psoriasis treatment.
Answer to the Reviewer: Thank you for your statement. An socio-economic status includes the place of residence which is included in Table 1. In my questionnaire, I asked patients about alcohol consumption, none of the 184 patients admitted addiction.
Q3. The definitions of PASI and BSA should be clearly explained in the manuscript so that readers from different fields could understand the research methods.
Answer to the Reviewer: Thank you for your statement. PASI and BSA definitions have been added to the manuscript.
Q4. Authors decided to compare the psoriasis severity before, after a ‘3-month treatment’. It was still unclear why authors used this treatment interval. Authors should provide clear information for the authors.
- Answer to the Reviewer: Thank you for your statement. The 3-month period is a period long enough to evaluate the first effect of treatment, improve quality of life. Many conventional and biological drugs are evaluated after a 3-month period.
Minor concerns
Q5. In Materials and Methods, using the percentage to present the data should be '.' instead of ','.
Answer to the Reviewer: Thank you for your statement. The text has been corrected as with note.
Q6. In Discussion, it was inappropriate to repeat the detailed results of this study. Authors should review more previous studies for relevant issues and provide the clear statements in Discussion.
Answer to the Reviewer: Thank you for your statement. The subject of this manuscript is innovative. The discussion highlighted the main results to compare with other studies. There are few articles in the available literature that could be compared to our results. These articles are included in manuscript.
Thank you for your evaluation
Reviewer 2 Report
Psoriatic patients with comorbidities such as hypertension, diabetes, lipid disorders, overweight/obesity as well as those study subjects who were smokers had poorer HRQOL and presented more severe psoriatic skin lesions
The presence of comorbidities, overweight/obesity and nicotine dependence had a negative effect on the outcome of conventional treatment with methotrexate, cyclosporine, retinoids.
The outcome of therapy with biological agents, however, was independent of each of the analyzed factors
Biological agents were highly effective in the treatment of psoriatic skin lesions despite the presence of comorbidities, whereas methotrexate was effective even if patients had coexisting hypertension
A limitation to the study may be a different number of patients treated with conventional therapies, i.e. methotrexate, cyclosporine, retinoids, and biological agents. Most patients were treated with methotrexate (n=79), and the number of patients treated with biological agents was much smaller (n=20). The study group differed with regard to their sex, age and duration of psoriasis

Author Response
Answer to the Reviewer 1:
Thank you for your evaluation. I would like to answer for your questions.
Q 1: Is very informative and complete. It is not necessary the use of acronymous in this section.
My answer to the Reviewer 1: Thank you for your statement. I have removed this acronymous from this section.
Q.2. Abstract and Keywords:
The explanation of the meaning of DLQI and PASI must be included before their first use. The summary is well done.
Keywords. They are well selected, but you must include plaque psoriasis , and conventional and biologic treatment.
My answer to the Reviewer 1: Thank you for your statement. The explanation of the meaning of DLQI and PASI has been included before their first use. Plaque psoriasis, conventional and biologic treatment have been added to keywords.
Thank you for your evaluation.
Reviewer 3 Report
Nicely written, although quite long with moderate repetition.
Was this a prospective study, or retrospective review? Did you lodge the protocol on an international Clinical Trials site?
In the materials and methods section, the basic demographics are given in actual patient numbers. It would be more helpful if this was ‘percentages’ with the actual numbers afterwards, e.g.’ conventional systemic drugs, i.e. methotrexate (43%, 79 patients), ciclosporin (xx%, 42 patients), etc’
In the methods section, you give results (i.e. the demographics, age, sex, BMI, etc). These are usually included in the results but this may be OK; however you should probably explain your methodology first (i.e. the explanation of how BMI was calculated comes after the description of the BMI in patients.
How was treatment allocation decided? Was there a risk that co-morbidities affected which treatment patients were offered? Were patients on more than one systemic treatment? Did you correct for difference in dosage? Did all patients complete 3/12 of treatment, or were there any dropouts? Were some patients just trialling a conventional systemic agent on route to a biologic?
Did the number of patients with co-morbidities change over the 3 months? i.e are we looking at the same groups of patients?
Table 2: why include decimal points on integers?
It might be helpful to add the number of patients in each treatment group to the heading of Table 2 an 3.
With so many statistical comparisons, there is a significant risk of one or more comparison reaching a p value of <0.05 purely by chance.
There is a fair amount of repetition of results between the text and tables.
Table 4 – please check percentages as 4/143 (9.3%) is incorrect. Also the total numbers of patients don’t add up you state, 83 had hypertension and 143 without = 226, but there should only be 184 patients. I assume this may be because patients had more than 1 therapy, but it would be inappropriate to include a patient more than once
The number of patients on biologics is too small to make any comment about differences between biologics and the other treatments.
Author Response
Answer to the Reviewer 2: Thank you for your evaluation. . I would like to answer for your questions.
Was this a prospective study, or retrospective review? Did you lodge the protocol on an international Clinical Trials site?
Answer to the Reviewer 2: Thank you for your statement. It was a retrospective review. This study was approved by the ethic committee of the Medical University of Lublin (No KE- 0254/154/2019).
In the materials and methods section, the basic demographics are given in actual patient numbers. It would be more helpful if this was ‘percentages’ with the actual numbers afterwards, e.g.’ conventional systemic drugs, i.e. methotrexate (43%, 79 patients), ciclosporin (xx%, 42 patients), etc’
Answer to the Reviewer 2: Thank you for your statement. In the materials and methods section, the basic demographics are given as a percentage.
In the methods section, you give results (i.e. the demographics, age, sex, BMI, etc). These are usually included in the results but this may be OK; however you should probably explain your methodology first (i.e. the explanation of how BMI was calculated comes after the description of the BMI in patients.
Answer to the Reviewer 2: Thank you for your statement. A suggested, appropriate changes have been made.
How was treatment allocation decided? Was there a risk that co-morbidities affected which treatment patients were offered? Were patients on more than one systemic treatment? Did you correct for difference in dosage? Did all patients complete 3/12 of treatment, or were there any dropouts? Were some patients just trialling a conventional systemic agent on route to a biologic?
Answer to the Reviewer 2: Thank you for your statement. The treatment was implemented in accordance with the current recommendations of the Polish Dermatological Society. PASI, BSA, DLQI, disease history, research results, and co-existing diseases are included. Psoriatic patients suffered from another comorbidities for a long- time. Patients received one systemic treatment. Taking difference dosages of medications have been added to limitations. All patients completed 3 months of treatment. No patient dropped out of treatment. Patients in this study did not test a conventional drug on their way to biological treatment.
Did the number of patients with co-morbidities change over the 3 months? i.e are we looking at the same groups of patients?
Answer to the Reviewer 2: Thank you for your statement. No. The number of patients wit co-morbidities did not change over the 3 months.
Table 2: Why include decimal points on integers? It might be helpful to add the number of patients in each treatment group to the heading of Table 2 and 3. With so many statistical comparisons, there is a significant risk of one or more comparison reaching a p value of <0.05 purely by chance. There is a fair amount of repetition of results between the text and tables.
Answer to the Reviewer 2: Thank you for your statement. Decimal points have been removed in tables. The number of patients in each treatment group have been added to the heading of table 2 and 3.
Table 4 – please check percentages as 4/143 (9.3%) is incorrect. Also the total numbers of patients don’t add up you state, 83 had hypertension and 143 without = 226, but there should only be 184 patients. I assume this may be because patients had more than 1 therapy, but it would be inappropriate to include a patient more than once. The number of patients on biologics is too small to make any comment about differences between biologics and the other treatments.
Answer to the Reviewer 2: Thank you for your statement. The error of percentage in table 4 has been corrected. 184 patients participated in this study in this manuscript. The number of 226 patients also took into account patients treated topical treatment who were excluded from study. The study concerns 184 patients with moderate or severe psoriasis. The results are statistically remained unchanged. The results presenting p values below 0.05 were considered as statistically significant. The error of the total numbers of patients has been corrected. Each patient was treated only 1 method of treatment.
Thank you for your evaluation
Round 2
Reviewer 1 Report
Thank you for the authors tried to answer the questions and revised the manuscript. I have reviewed the author’s responses one by one including other reviewers. However, there were still a few concerns I would like the author to explain in more detail:
Q1. Authors stated that “None of the demographic factors affected either DLQI or PASI score, thus the whole studied group underwent non-parametric testing." in p.4 line 104. However, the statements were inconsistent with the “The study group differed with regard to their sex, age and, duration of psoriasis and different dosages of treatment” in Limitations (p15 line 363). I strongly recommend authors to present the relevant data in Tables.
Q2. The clear explanation about the 3-month treatment interval should be addressed in the manuscript.
Q3. The author must review more previous studies and provide clear Discussion in the manuscript.
Author Response
Thank you for the authors tried to answer the questions and revised the manuscript. I have reviewed the author’s responses one by one including other reviewers. However, there were still a few concerns I would like the author to explain in more detail:
Q1. Authors stated that “None of the demographic factors affected either DLQI or PASI score, thus the whole studied group underwent non-parametric testing." in p.4 line 104. However, the statements were inconsistent with the “The study group differed with regard to their sex, age and, duration of psoriasis and different dosages of treatment” in Limitations (p15 line 363). I strongly recommend authors to present the relevant data in Tables.
Q2. The clear explanation about the 3-month treatment interval should be addressed in the manuscript.
Q3. The author must review more previous studies and provide clear Discussion in the manuscript.
Thank you for your useful evaluation. I would like to answer to all your questions exhaustively. Your remarks are very important for us.
Q1. Authors stated that “None of the demographic factors affected either DLQI or PASI score, thus the whole studied group underwent non-parametric testing." in p.4 line 104. However, the statements were inconsistent with the “The study group differed with regard to their sex, age and, duration of psoriasis and different dosages of treatment” in Limitations (p15 line 363). I strongly recommend authors to present the relevant data in Tables.
My Answer: Thank you for your statement. Thank you for your useful evaluation. Comparisons were made only within one group of medication ( for example methotrexate alone, cyclosporine alone, retinoids alone, biological drugs alone) . Comparisons and analysis were not made between drug groups ( for example cyclosporine /retinoids or cyclosporine/biological drugs). There were no comparisons between the subgroups, which eliminates the influence of clinical-demographic characteristics on the results obtained. Work limitations refer to the fact that EACH of the subgroups should be similar to the others in terms of the factors mentioned ( their sex, age, duration of psoriasis and different dosages of treatment), which would allow the most objective conclusions and results obtained for "all applied therapies" to be drawn, where the value of each therapy would be similar. Your remark is very important, interesting and inspiring for us. In future studies we are going to perform a pairwise comparison of drugs.
In ,, limitation section’’ we have been added information that ,, The EACH of study group differed with regard to their sex, age, duration of psoriasis and different dosages of treatment’’.
According to your previous suggestion clear information has been added previous to the section matherial and methods :
,, Statistical analysis was conducted with the use of the MedCalc computer software, version 18.2.1 (MedCalc, Belgium). Data distribution in the studied population was examined by the Shapiro-Wilk test. All the studied parameters demonstrated abnormal distribution of the data (p<0.05)therefore, non-parametric tests were applied in the further analysis. Multiple regression analysis for the clinical-demographic features of the studied individuals was applied to avoid the potential confounding effects on the results. None of the demographic factors affected either DLQI or PASI score, thus the whole studied group underwent non-parametric testing. Values of the studied factors in the subgroups of patients treated with different therapy regimens were expressed by the median (Me) and interquartile range (IQR). Differences in the values of the studied parameters among the patients in regard to the presence of comorbidities, BMI score and smoking status were tested by the Mann-Whitney U test. The results presenting p values below 0.05 were considered as statistically significant’’.
Thank you very much for you useful evaluation and interesting remarks. Thanks for this remark we have an idea for a new research comparing drug groups.
Q2. The clear explanation about the 3-month treatment interval should be addressed in the manuscript.
- My answer: Thank you for your statement. The 3-month period is a period long enough to evaluate the first effect of treatment and improvement in quality of life. Many conventional and biological drugs are evaluated after a 3-month period. For example, in a large Anzengruber’s study (1264 patients) which was cited in discussion in our manuscript, the effect of the treatment was evaluated after 3 months. The clear explanation about the 3-month period of treatment has been addressed in the manuscript ( in the section material and methods, characteristics of the studied group). Thank you for your useful evaluation.
Q3. The author must review more previous studies and provide clear Discussion in the manuscript.
My answer: Thank you for your statement. There are few articles in the available literature that could be compared to our results. As suggested by the reviewer, we have added a few more previous studies from the literature:
-Michalska- Bańkowska A, Grabarek B, Wcisło- Dziadecka D, Gola J. The impact of diabetes and metabolic syndromes to the effectiveness of cyclosporine a pharmacotherapy in psoriatic patients. Dermatol Ther 2019; 32 (3): 1-5
-Hayran Y, Yalcin B. Smoking habits amongst patients with psoriasis and the effect of smoking on clinical and treatment-associated characteristics: A cross- sectional study. Int J Clin Pract 2021; 75 (2) 1-22
- Prignano F, Pescitelli L, Ricceri F, Lotti T. Retrospective analysis of systemic treatments for psoriasis patients attending a Psocare center in Florence. Relevance of biological drugs use and comorbidities. J Eur Acad Dermatol Venereol. 2010; 24( 5): 555-560
Thanks to new literature, the discussion can be expanded. Thank you very much for your evaluation.